# Physiological and Biochemical Responses of Ungrafted and Grafted Bell Pepper Plants (*Capsicum annuum* L. var. *grossum* (L.) Sendtn.) Grown under Moderate Salt Stress

**DOI:** 10.3390/plants10020314

**Published:** 2021-02-06

**Authors:** Nina Kacjan Maršić, Petra Štolfa, Dominik Vodnik, Katarina Košmelj, Maja Mikulič-Petkovšek, Bojka Kump, Rajko Vidrih, Doris Kokalj, Saša Piskernik, Blaz Ferjančič, Maja Dragutinović, Robert Veberič, Metka Hudina, Helena Šircelj

**Affiliations:** Biotechnical Faculty, University of Ljubljana, 1000 Ljubljana, Slovenia; petra5stolfa@gmail.com (P.Š.); dominik.vodnik@bf.uni-lj.si (D.V.); katarina.kosmelj@bf.uni-lj.si (K.K.); maja.mikulic-petkovsek@bf.uni-lj.si (M.M.-P.); bojka.kump@bf.uni-lj.si (B.K.); rajko.vidrih@bf.uni-lj.si (R.V.); doris.kokalj@bf.uni-lj.si (D.K.); sasa.piskernik@bf.uni-lj.si (S.P.); blaz.ferjancic@bf.uni-lj.si (B.F.); maja.dragutinovic@bf.uni-lj.si (M.D.); robert.veberic@bf.uni-lj.si (R.V.); metka.hudina@bf.uni-lj.si (M.H.); helena.sircelj@bf.uni-lj.si (H.Š.)

**Keywords:** grafting, ascorbic acid, phenolics, NaCl, photosynthetic traits, proline, *Capsicum annuum* L.

## Abstract

The response of grafted bell pepper plants (*Capsicum annuum* L. var. *grossum* (L.) Sendtn.) to salt stress was investigated by analyzing the photosynthetic traits and mineral content of the plants and the metabolic composition of the fruit. The bell pepper variety “Vedrana” was grafted onto the salt-tolerant rootstock “Rocal F1” and grown at two salinities (20 mM and 40 mM NaCl) and control (0 mM NaCl) during the spring–summer period. On a physiological level, similar stomatal restriction of photosynthesis in grafted and ungrafted plants indicated that grafting did not alleviate water balance disturbances under increased salt exposure. Measurements of midday water potential did not show improved water status of grafted plants. The similar metabolic changes in grafted and ungrafted plants were also reflected in similarly reduced fruit yields. Thus, this grafting did not reduce the risk of ionic and osmotic imbalance in pepper plants grown under moderate salt treatment. Changes in the biochemical profiles of the pepper fruit were seen for both added-salt treatments. The fruit phenolic compounds were affected by rootstock mediation, although only for the July harvest, where total phenolics content increased with 40 mM NaCl treatment. Fruit ascorbic acid content increased with the duration of salt stress, without the mediation of the rootstock. The high salt dependence of this quality trait in pepper fruit appears to lead to more limited rootstock mediation effects.

## 1. Introduction

Sweet pepper (*Capsicum annuum* L.) is a widespread and popular crop. When grown in arid and semi-arid areas, pepper often encounters soil salinity. [1,2]. Soil is considered to be saline when the electric conductivity of the soil solution reaches 4 dS·m^−1^ (equivalent to 40 mM NaCl). The salinity threshold level of pepper plants is 1.5 dS·m^−1^, thus pepper is considered to be moderately salt-sensitive [3]. An increase in soil salinity induces osmotic, ionic, and oxidative stress in the plants [4,5]. This can lead to reduced ion uptake and greatly increased outflow of water and ions (e.g., K^+^) in plant cells, which results in water and nutritional imbalance [6]. Moderate salinity levels mainly induce osmotic stress, while high salt concentrations also induce toxicity from Na^+^ and Cl^–^ ions [7]. 

Growth inhibition occurs in many plants when they are exposed to salinity, and this is often associated with decreased photosynthetic capacity, which can be due to stomatal or nonstomatal restrictions [8]. Because photosynthetic CO_2_ fixation is limited under saline stress, the rate of light energy absorption by photosynthetic pigments exceeds the rate of its consumption in chloroplasts [9], which can accelerate photosynthetic damage to photosystem (PS)II through the formation of reactive oxygen species [10]. 

Many different pathways have been proposed to cooperate in the protection of the photosynthetic apparatus from photooxidative stress, such as photorespiration, xanthophyll-cycle-dependent energy dissipation, cyclic electron flow through both PSI and PSII, and the antioxidant system [11,12]. 

Plants can cope with hyperosmotic stress by increasing their endogenous concentrations of osmotically active metabolites, such as proline [13], and by maintaining or increasing in tissues K^+^ (high-affinity potassium transporters). Shoot water deficits are mitigated by stomatal regulation for increased water use efficiency, increased wall extensibility, and increased root hydraulic conductance. Adverse effects of high Na^+^ and Cl^−^ concentrations in the leaves are reduced by vacuolar compartmentation, tissue distribution, and organ-level partitioning [14]. 

A promising perspective for vegetable production in areas with high soil salt levels is the grafting of commercial varieties onto salt-tolerant rootstock [15,16,17,18]. The mechanism of salinity resistance in grafted plants are various pepper plants grafted onto tolerant rootstock (e.g., *Capsicum chinense* Jacq., *Capsicum baccatum* L.) can achieve greater plant productivity due to limiting of Cl^–^ transport to the leaves by the rootstock and reduction of the Na^+^ load in the roots and leaves. This allows the uptake of other cations (i.e., K^+^, Ca^2+^, Mg^2+^), for lower osmotic potential at lower energy cost [16]. However, these effects cannot be generalized due to the great specificity of scion/rootstock interactions [16,19,20,21]. In addition, currently available commercial rootstock lines of pepper offer modest benefit [22]. 

The quality characteristics of pepper fruit in terms of grafting have been assessed in a few studies [23,24,25]. These have shown a strong influence of the rootstock on the differences in morphometric characteristics, and biochemical composition of pepper fruit [15,23], with many contradictory results [26,27,28,29]. Soluble sugars and total acid concentration in pepper fruits generally do not seem to be strongly affected by grafting, although some contrary results can also be found in the literature [24,30,31]. As functional compounds, the polyphenols content of pepper fruit has been shown not to be affected by grafting [25,27,32]. The ascorbic acid in pepper fruit is of considerable importance, although again there are contradictory data in the studies currently available, in terms of the variations of ascorbic acid content in response to grafting [25,29,31,32]. It has also been suggested that high genotype dependence of the ascorbic acid content in pepper scions impairs the more limited effects of the rootstock [23,33].

The majority of published salinity and grafting studies with bell pepper have been conducted to assess the tolerance of grafted plants to NaCl over short durations of moderate and severe salt stress (10–30 days) [16,17,34]. To the best of our knowledge, there have been no studies on the physiological and biochemical stress indicators or with regard to the quality of fruit yield for grafted versus ungrafted bell pepper plants grown under moderate salt stress during the summer period. 

To better understand the relationships of technological factors (e.g., grafting) combined with abiotic stress (e.g., salinity), the present study investigated the responses of ungrafted and grafted pepper plants to different salt concentrations during the summer growing season. These responses included (1) fruit yield of “Vedrana” bell pepper plants without and with grafting on “Rocal” rootstock; (2) physiological and biochemical parameters as salt stress indicators in these bell pepper plants; and (3) how the effects of grafting and salinity translate into the biochemical composition of the pepper fruit. Analysis of ungrafted and grafted pepper plants thus included gas exchange parameters, chlorophyll fluorescence, photosynthetic pigments, proline concentration, and mineral composition for the pepper plant leaves, and the metabolic profile of the pepper fruit.

## 2. Results

### 2.1. Fruit Yield

The data for fruit yields are expressed as the mean mass of marketable fruit per plant (kg·plant^−1^) and the proportion (%) of fruit affected by blossom-end rot, as presented in Table 1. A linear mixed model was implemented for the marketable yield. These data show significant decreases for both ungrafted and grafted plants for the marketable fruit yield and the plant biomass for increased salinity (*p* < 0.0335; *p* < 0.0322), with greater decreases with the increased salt stress. However, there was no significant effect of grafting itself on fruit yield or plant biomass, but on the blossom end rot symptoms appearing in pepper fruits, the effect of grafting itself was significant (*p* < 0.0001). The significantly fewer blossom end rot symptoms were appeared in fruits of grafted plants compared to the ungrafted control plants from the highest salt stress condition (40 mM NaCl).

### 2.2. Water Relations

For the reductions seen for midday leaf water potential (ψ) in ungrafted and grafted plants, salinity (i.e., salt stress) and date of measurements proved to be highly significant factors (*p* = 0.0002, *p* < 0.0001, respectively). Under the salt stress conditions of 20 mM NaCl and 40 mM NaCl, the ψ at all three sampling dates was significantly reduced compared to the control (0 mM NaCl). The greatest differences between these salt treatments were at the mid-duration in August, when the ψ values were −0.75 ± 0.04 MPa, −0.95 ± 0.06 MPa, and −1.25 ± 0.08 MPa for 0 mM, 20 mM, and 40 mM NaCl, respectively. These differences were least prominent for the early treatment time in July. However, the grafting itself did not have any significant effects on ψ for any of the sampling periods (Figure 1).

### 2.3. Photosynthetic Traits

The effects of this salt stress were reflected in reduced stomatal conductances (g*_s_*), for both ungrafted and grafted plants (Table 2). Here, significant impacts of increased salt stress (*p* = 0.0319) and date of measurements (*p* < 0.0001) were seen, while the effects of the grafting did not show statistical significance (*p* = 0.0913). These differences in g*_s_* were similarly paralleled by the reduced transpiration rates (*E*) (Table 2), where the duration of treatment had a significant impact (*p* < 0.0001), although grafting (*p* = 0.080) and salt stress (*p* = 0.088) had no significant effects here. The average *E* of the ungrafted and grafted plants was higher in July than in August and September. 

For the photosynthesis measure of net CO_2_ assimilation rate (Pn), there were again significant impacts of increased salt stress (*p* = 0.0195) and salt stress duration (*p* < 0.0001), and also the interaction between salinity and grafting was of marginal significance (*p* = 0.0531). In July, there were no effects of salt stress or grafting on the photosynthetic traits of these pepper plants (Table 2). With the longer treatments in August and September, Pn decreased with increasing salt stress in the ungrafted plants, while the grafted plants only showed decreased Pn under the higher salt stress (40 mM NaCl). As the lower Pn values corresponded to reduced *gs* (i.e., stomatal conductance), this suggested stomatal limitations on photosynthesis. 

Compared to the control (0 mM NaCl), the maximum quantum use efficiency of PSII in the dark-adapted state (Fv/Fm) was significantly affected only by the duration of the salt stress (*p* = 0.0305) (Table 3). This was reflected in the reduction of Fv/Fm under salt stress only for the longer treatments in September, with no significant differences between ungrafted and grafted plants. For the effective quantum use efficiency of PSII in the light-adapted state (Fv′/Fm′) there were significant impacts of increased salt stress (*p* = 0.0240), salt stress duration (*p* = 0.0096), and the two-factor interactions between salinity and grafting (*p* = 0.0218). Similar to Pn, there were significant reductions of Fv′/Fm′ under the increased salt stress conditions compared to the control (0 mM NaCl) in the ungrafted plants in August. For the grafted plants, Fv′/Fm′ reduction occurred only under the higher salt stress (40 mM NaCl) throughout the duration of the treatments. The photochemical quenching coefficient (q_P_) was significantly influenced only by the salt stress duration (*p* = 0.0001), as a significant decrease in q_P_ over time for these treatments for both the ungrafted and grafted plants (Table 3). 

### 2.4. Photosynthetic Pigment Characteristics and Xanthophyll Cycle Components

The photosynthetic pigments in the pepper leaves were analyzed in July and September 2018. Compared to the control (0 mM NaCl), the increased salt stress did not affect the total chlorophyll content (Chl_a+b_) in these pepper leaves, although there was a significant impact of salt stress duration on Chl_a+b_ (*p* = 0.0050). This was reflected in decreased Chl_a+b_ in the pepper leaves with the higher salt stress (40 mM NaCl), which was more evident with the longer duration of the treatments in September. At that time, the ungrafted plants had lower Chl_a+b_ compared to the grafted plants, with these differences reaching marginal significance (*p* = 0.060) (Table 4).

The total xanthophyll cycle pool of violaxanthin plus antheraxanthin plus zeaxanthin (VAZ) decreased significantly with increasing salt stress (*p* = 0.0017). The de-epoxidation state of the xanthophyll cycle (EOS), which was calculated as the (A plus Z)/(V plus A plus Z) ratio, showed significant decreases with increased salt stress early on in July (*p* = 0.0001) and then seen for the longer treatments in September (Table 4).

### 2.5. Proline Levels

The proline levels in the leaves of these ungrafted and grafted pepper plants are shown in Figure 2. Here, a linear mixed model was implemented using the log-values of the proline levels (Figure 2). There were significant effects of salinity and treatment duration, and also of the interaction between these (S × D) (*p* = 0.0004) and between treatment duration and grafting, (D × G) (*p* = 0.0149) (Figure 2). The differences between the ungrafted and grafted plants showed significance only with the longer treatments in September for both of the salt stress treatments with the grafted plants showing leaf proline levels significantly reduced by about 40% compared to the ungrafted control plants for both of the salt stress conditions (20 mM, 40 mM NaCl).

### 2.6. Mineral Contents

The levels of Na^+^, Cl^–^, K^+^, Ca^2+^, and Mg^2+^ in the pepper leaves were analyzed following the longer (76 days) salt stress treatments in September (Table 5). 

Here, although there were no significant differences between the ungrafted and grafted plants, there was significant impact of salt stress on the leaf Na^+^ levels (*p* = 0.016), with the combined data (i.e., ungrafted plus grafted) increasing from Na^+^ levels of 0.015 ± 0.006 mg·g^–1^ FW at 0 mM NaCl, to 0.41 ± 0.21 mg·g^–1^ FW and 1.07 ± 0.47 mg·g^–1^ FW at 20 mM and 40 mM NaCl, respectively. 

For the salt stress increases in leaf Cl^–^ levels, the difference between ungrafted and grafted plants was significant for the higher salt stress (40 mM NaCl; *p* = 0.0002), reflecting the higher Cl^–^ levels in grafted plants (5.61 ± 0.62 mg·g^−1^ FW) compared to ungrafted plants (4.73 ± 0.41 mg·g^−1^ FW). 

While there were no significant effects of salt stress and grafting on the leaf K^+^ levels, a significant impact was seen for salt stress on leaf Ca^2+^ and Mg^2+^ levels (*p* = 0.0067, *p* = 0.0428, respectively). These were seen as significant increases for these minerals under the greater salt stress, for both the ungrafted and grafted plants. 

### 2.7. Biochemical Parameters of the Pepper Fruit

In addition to analysis of the effects on the total marketable fruit yield of both the salt stress and the grafting, a range of biochemical parameters of the pepper fruit were investigated. Data for the total sugars, organic acids, ascorbic acid, and phenolics are presented in Table 6.

The total sugar content represents the sum of the sucrose, glucose, and fructose contained in the pepper fruit. None of the factors examined had any significant influences on the total sugars here. 

The total organic acids represent the sum of the malic, citric, and fumaric acids in the pepper fruit. Here, the two-factorial interactions between salt stress and duration of treatment and between salt stress and grafting were significant and marginally significant, respectively (*p* = 0.0001, *p* = 0.0559). These data showed that for the longer treatment to September, the organic acids tended to be higher in the fruit of the ungrafted plants than the grafted plants. However, these potential differences did not show significance, because of the small number of fruit samples analyzed, due to a shortage of fruit during this salt stress in September that was caused by blossom-end rot. 

The ascorbic acid content increased significantly (*p* < 0.0001) with the duration of the salt stress, thus showing higher levels in September compared to July. These increases with increased salt stress in September were also significant (*p* = 0.0086). The two-factor interaction between salt stress (salinity) and duration of treatment (July versus September) was also significant (*p* = 0.009), while the grafting treatment had no effect on the ascorbic acid contents of the fruit within these salt stress conditions, for both of the two sampling dates. 

We identified and quantified 16 phenolics in the pepper fruit, as hydroxycinnamic acids and flavonoids (i.e., flavonols, flavones). These total phenolics in the fruit decreased significantly with the duration of treatment (July versus September; *p* < 0.0001) (Table 6). ANOVA showed that the three-factor interaction (S × G × D) was also significant (*p* = 0.0048), which means that the increased salt stress in July led to decreased total phenolics in the fruit of the ungrafted plants. Conversely, in the fruit of the grafted plants in July, there was an increase in the total phenolics with the higher salt stress (40 mM NaCl), which resulted in a significant difference between the grafting treatments (*p* = 0.0048).

## 3. Discussion

### 3.1. Physiological and Biochemical Parameters of Pepper Plants

In the present study, the responses of the ungrafted and grafted pepper plants to the salt stress induced by these salinity treatments were evaluated by analyzing some of the physiological and biochemical parameters in the leaves and fruit over increased treatment duration. 

The lower accumulation of plant biomass and the reduced fruit yield associated with these treatments will probably have arisen as a consequence of the negative effects of salt stress, which is often a result of the inhibition of photosynthesis [35]. In general, a Pn reduction might be due to lower intercellular CO_2_ concentrations in the leaves as a result of stomatal closure, or because of nonstomatal factors, including conversion and dissipation of the photon energy into heat through the xanthophyll cycle, or degradation of photosynthetic pigments due to severe stress [36]. In the present study, with the longer duration of salt stress, Pn in the ungrafted plants always showed a decrease with increased salt stress, whereas Pn in the grafted plants decreased only under the higher salt stress. At the same time, no differences between ungrafted and grafted plants were seen for the increasing salt stress and its duration in terms of the reduction in stomatal conductance (g*_s_*) and transpiration (*E*). This indicated that the stomatal restriction of photosynthesis [37] was similar in the ungrafted and grafted plants. As the stomatal function reflects the water status of the plants, we can assume that the grafting did not reduce the disturbance in the water balance under the increased salt stress. These data do not support previous studies that have shown that grafting can improve the water status of pepper plants [38,39,40]. The rootstock used in this study failed to obtain scions that showed better physiological performance and consequently higher yield, although it underwent a rigorous screening program for salt tolerance [41]. It has been reported that some of the salt-tolerant rootstocks were tested for many years under real salinity field conditions and showed higher yields than ungrafted plants or other commercial rootstocks tested [16]. Therefore, further studies with different salt-tolerant rootstocks are needed in the future to confirm the tolerance of grafted plants in prolonged salt stress conditions.

The analyses of the photosynthetic pigments and fluorescence measurements indicated that Pn was also affected by nonstomatal limitations. There was a decrease in Chl_a+b_ levels with increasing duration of the salt stress. The other effect that can reduce Pn that was also monitored here was photoinhibition [12,39,42]; in the ungrafted plants, Fv′/Fm′ decreased significantly with increased salt stress, whereas in grafted plants, it decreased only under the higher salt stress (40 mM NaCl). This decrease can mainly be attributed to the efficiency of the excitation energy capture by the open PSII reaction centers [43], which suggests that in the grafted plants, the photoinhibition of PSII was triggered with a delay. The de-epoxidation state ratio (EOS) did not indicate the involvement of the xanthophyll cycle pigments in this Pn reduction. 

Comparisons of the data for the measurements carried out for the different treatment durations through the season suggest that the mitigating effects of grafting were seen most strongly when these pepper plants were exposed to unfavorable growth conditions, which will be combined with the increased duration of the primary stressor, salinity. The reduction in Pn and Fv′/Fm′ in August was most likely also due to the high temperatures and low humidity, which will affect leaf physiology. From this, it can be concluded that grafted plants appear to maintain higher stomatal conductivity and photochemical efficiency under these conditions.

The underlying mechanism here will not be related to the mineral contents of the leaves because no significant differences between the ungrafted and grafted plants were seen for the K^+^ and Na^+^ levels in the salt-exposed pepper plants. This indicates that in the present study, the grafting did not reduce the risk of ionic and osmotic imbalance. Studies with grafted plants that have reported this effect have proposed salt exclusion from the shoots and its retention in the roots [38], or compartmentalization of the salt ions in the cell vacuole [26]. The ability of some rootstocks to retain Na^+^ ions in their roots is genotype-specific, as has been reported for rootstock of peppers, cucumbers, and tomatoes [4,20,44,45]. However, this was not confirmed in the present study because no measurements of Na^+^ concentration were carried out for the roots of these pepper plants. 

Better maintenance of K^+^ homeostasis in plant tissue is another salt tolerance mechanism that grafting can affect [26]. The K^+^ content in leaves of pepper plants in our study is in agreement with a previous study [20] where no significant effect of salt treatment on K+ content in leaves of pepper plants was found when salt stress was induced by adding NaCl to the common nutrient solution. K^+^ assists in the cation–anion balance, osmoregulation, and water movement and is essential for plant acclimation to biotic and abiotic stress [46]. The retention of K^+^ ions in the cells of the roots and leaves has been shown to be a selection criterion between salt-tolerant and salt-sensitive varieties. Although the “Rocal” rootstock used in this study was selected as salt-tolerant, there were no significant effects of grafting on the K^+^ and Na^+^ levels in the leaves of these pepper plants under moderate salt exposure during the summer growing season. These results are in agreement with previous findings reported by Aktas et al. [5], who also observed no differences in leaf K^+^ content between control and NaCl-stressed pepper plants of genotypes classified as "tolerant." The salt stress in the present study, induced by the addition of NaCl to the common nutrient solution presumably resulted in competition between Na^+^ and the cations already present in the nutrient solution, thereby attenuating the salt stress. Indeed, it is known that the addition of external Ca^2+^ and K^+^ can significantly mitigate salinity stress symptoms in many species [47]. In our study, cation competition probably reduced salt stress to the extent that tolerance of grafted plants to salinity could not be expressed. Therefore, in future studies, modified solutions with lower cation concentrations of macronutrients should be used in salt stress treatments in addition to the usual nutrient solution to investigate whether the reduction of salt stress due to competitive cation effects has an influence on the growth and yield of grafted and ungrafted pepper plants. 

Although the K^+^ levels in the leaves of these ungrafted and grafted plants indicated similar salt stress responses, the Cl^–^ levels showed the opposite. Here, the Cl^–^ levels significantly increased under the higher salt stress, by about 27.8-fold in ungrafted plants and 37.4-fold in grafted plants. This resulted in significant differences in Cl^–^ concentration in these treated plants and reflected the inability of the plant/ rootstock to limit the transport of toxic Cl^–^ ions to the shoot. Similar data on the lack of rootstock retention of Cl^–^ ions have been obtained in other studies on salt-tolerant grafted pepper plants [16,17] and salt-tolerant grafted tomato plants [24]. Navarro et al. [2] also reported negative effects of salinity on pepper growth, and they concluded that the yield reduction induced by salt stress can be linked to the toxic effects of Cl^–^ accumulation in the plant tissues. Indeed, this might be one of the reasons for the yield reduction under the salt stress in the present study. It is known that the Cl^−^ concentration in the shoot of non-halophyte plants varies greatly, ranging from 1 mg·g^−1^ to 20 mg·g^−1^ DW [48,49], which means that the Cl^−^ concentrations in this study, which is expressed in DW amounted to 35 mg·g^−1^ and 41 mg·g^−1^, indeed reached toxic amounts. In the future, whole-plant Cl^−^ content analysis may be included in studies to provide data for a better understanding of the salt stress mechanism that will enable grafted pepper plants to overcome salt stress problems during the growing season. Proline is one of the compounds that can accumulate in plant tissues, most frequently as an osmolyte or protective substance under unfavorable environmental conditions, such as drought and salt stress [6,50]. Salt-induced proline accumulation was also evident in the present study. Under these prolonged moderate salt stress conditions, the ungrafted plants showed higher leaf proline levels than the grafted plants, but at the same time, they had lower effective quantum efficiency (Fv′/Fm′), which indicated higher stress intensity. However, as the rootstock did not contribute to the exclusion of salt from the shoots, a similar level of osmotic disturbance would be expected in the leaves of both of these plant groups. Differences in leaf proline levels might be due to different involvement of this compatible organic solute in osmotic adjustment, which is also favored by salt ions deposited in the vacuole [14]. Here, the high Cl^−^ levels in the leaves of the grafted plants might have an important role. As the ungrafted and grafted plants showed similar water status throughout the experimental period and had similar responses of the stomata under different conditions, it can be excluded that the differences in the proline levels are a result of a specific response of plants to water deficit (e.g., high vapor pressure deficit; see Grossiord et al. [51]). 

### 3.2. Fruit Quality

In addition to providing resistance to soil-borne pathogens [52], the main objective of vegetable grafting is an improvement of tolerance to abiotic stress [15] thus promoting increased yields, although often at the expense of fruit quality [33]. It is therefore of utmost importance to understand the effects of grafting on the fruit quality parameters and to understand and define the mechanisms involved [23]. 

To the best of our knowledge, there are no reports in the literature of changes in the biochemical profiles of pepper fruit induced by salinity and grafting, with treatment durations continuing across different harvest periods (i.e., July to September). In the present study, the data showed no influence of salt stress on the total sugars content in the pepper fruit, and no differences in the total sugars content between ungrafted and grafted plants, under both the short treatment duration for fruit harvested in July and the longer treatment to September. The total acidity showed similar results, although, for the fruit harvested in September, the total acidity of the grafted plants tended to be lower than that of the ungrafted plants, even if the differences did not show significance. The results of the fruit acidity might be associated with less severe salt stress in the grafted plants because total sugar content and titratable acidity in the pepper fruit increase as water loss of the plant increases, due to osmotic drought caused by salt stress [53]. 

Among the functional compounds reported to be influenced by salinity and grafting [15], ascorbic acid and polyphenols were analyzed in this study. In the literature, contradictory data have been reported regarding variations in the ascorbic acid content in response to grafting [25,29,30,31,32]. The present study showed no significant impact of grafting on ascorbic acid content for both the short-term and the long-term treatments. This is consistent with data reported by Sánchez-Torres et al. [32], who showed that grafting of two pepper varieties onto two different rootstocks did not alter the ascorbic acid contents in the pepper fruit. In the present study, the salt stress influenced the ascorbic acid content only with the long-term treatment in September, which resulted in ascorbic acid increases with increased salt stress under both of these grafting conditions. Effects of salt stress on ascorbic acid content have been shown in some studies, although the data are again contradictory. For example, in green pepper, osmotic stress induced by low irrigation frequency strongly increased the ascorbic acid content (by 23%), while an increase in salinity from 0 mM to 30 mM NaCl resulted in a decrease in ascorbic acid content [53]. In contrast, in cherry tomato fruit, increased salt in the nutrient solution resulted in increased ascorbic acid. This suggested that the increase was a consequence of the increase in fruit dry matter due to the different salinity conditions and the activation of specific metabolic pathways in tomato plants under high salt stress conditions [54].

Polyphenols are phytochemicals that contribute to antioxidant activities in plants, and they are present in pepper fruit at moderate to high levels [55]. The synthesis of phenolics has been described as actively involved in neutralization of free radicals as a response to oxidative stress caused by abiotic factors [15,33]. The high pressure liquid chromatography HPLC–mass spectrometry analysis of the pepper fruit in the present study showed that the phenolic profile mainly consisted of hydroxycinnamic acid and two types of flavonoids—flavones (for the most part) and flavonols. Here, the phenolic content in pepper fruit decreased with the duration of the salt stress. This higher total phenolics in fruit from the short-term treatments compared to those from the long-term treatments might be a further indicator of the stress caused by environmental factors compared to the fruit from the long-term treatments because stress conditions such as high solar radiation and temperature induce an accumulation of phenolics [29,56]. We obtained similar data in our previous study, in which the symptoms of salinity were differently expressed in tomato plants that were stressed in mid-summer or late summer [57]. 

Data on variations in the phenolics in grafted vegetable fruit are contradictory. Most previous studies have investigated biochemical responses of plants to salt treatment, while few have investigated instead the biochemical responses of fruit. For instance, Koleška et al. [58] reported that moderate salinity led to their greatest reductions in flavonoids in ungrafted tomato plants, while at the same time, it caused the highest increase in flavonoids in the fruit. This suggested that the phenolics are transported via the phloem from the leaves to the fruit, where their antioxidant protection occurs. Similarly, López-Marín et al. [30] reported a decrease in total phenolics in “Herminio” pepper plants, but only when “Creonte” rootstock was used. In contrast, López-Serrano et al [34] reported significant increases in total phenolics in grafted pepper plants under salinity treatment, which coincided with stimulation of antioxidant capacity because phenolic compounds are reported to help prevent the formation of reactive oxygen species and protect the photosynthetic apparatus [59]. In the present study, for the short-term treatments in July, the grafting led to significant increases in total phenolics content in these pepper fruit under the higher salt stress (40 mM NaCl), compared to the ungrafted plants. This might indicate that higher salt stress in July, accompanied by harsh environmental conditions, triggers a defense mechanism in grafted plants that includes an increase in phenolics and has a protective role against ion-induced oxidative stress [34]. With the longer-term treatments in September, however, there were no significant differences in the total phenolic content between the fruit of the ungrafted and grafted plants. 

## 4. Materials and Methods

### 4.1. Plant Material and Growth Conditions

This study was carried out in spring and summer of 2018, in an unheated greenhouse in an experimental field of the Biotechnical Faculty of the University of Ljubljana (Slovenia; 46°2′ N; 14°28′ E; 298 m a.s.l.). Ungrafted “Vedrana” bell pepper plants (Enza Zaden, Enkhuizen, Netherland) were used as the control plants. For the grafted plants, “Vedrana” was used as the scion and “Rocal” (Esasem, Verona, Italy) as the rootstock. Cultivar “Vedrana” belongs to white bell pepper (*Capsicum annuum* L. var. *grossum* (L.) Sendtn.), which is an important crop in southeastern Europe and in other European regions [60,61]. It is used either for domestic consumption or export. The “Rocal” rootstock is registered as rootstock with a strong root system that is cold and salt-tolerant (Esasem, 2018) [41]. The seeds of the scions were sown on 7 March 2018 in 84-cell polystyrene trays (cell volume 35 mL) that were filled with a peat-based substrate (Neuhaus N3; Humko, Slovenia), and the seeds of the rootstock were similarly sown three days later. After seven weeks (24 April 2018), the scions were grafted onto the rootstock using the tube grafting method [22]. Briefly, the shoot tip of the rootstock was cut off below the cotyledons at an angle of 45°, with the same for the scion above the cotyledons. The grafting position was fixed firmly with a tubular silicone clip (1.5 mm hole diameter, JT Agro, Rome, Italy). During the formation of callus, plants were placed in the tunnel for acclimatization, covered with polyethylene (PE) mulch (0.18 mm) and a shaded screen. The temperature in the tunnel was between 20 °C and 25 °C, relative humidity was between 60% to 90%. For the first five days after grafting, the plants were completely shaded, then shaded screen was removed, thus plants were grown under PE mulch and grown under normal day/light regime. The salt stress treatments were based on 20 mM and 40 mM NaCl, to mimic the growing conditions in arid and semiarid regions where bell pepper is often cultivated [62].

For the hydroponic experiments, four weeks after the grafting, the root systems of the plants were washed to clean off the substrate and then transplanted into 6-L polyethylene pots filled with a 2:1 (*v*/*v*) mixture of perlite (particle size, 3–5 mm) and rock wool flocks. The hydroponic pot-drip system was used (Wilma; Nutriculture, Lancashire, UK), which consisted of pots and 85-L plastic tanks (120 cm × 60 cm). Each tank was covered with a plastic tray on which six plants (three ungrafted, three grafted) were placed randomly in rows 50 cm apart with 25 cm between the pots in each row (plant density, 6.67 plants m^−2^). 

The standard nutrient solution for plant watering contained the following ionic composition: 14.0 mM NO_3_^–^, 1.0 mM NH_4_^+^, 1.0 mM H_2_PO_4_^–^, 6.0 mM K^+^, 4.0 mM Ca^2+^, 2.0 mM SO_4_^2–^, 2.0 mM Mg^2+^, 2.0 mM Na^+^, 1.8 mM Cl^–^, 10 µM Mn^2+^, 5.0 µM Zn^2+^, 30 µM B^3+^, 15 µM Fe^3+^, 0.75 µM Cu^2+^, and 0.5 µM Mo^6+^. The pH and electrical conductivity of the nutrient solution were 6.5 dS·m^–1^ and 2 dS·m^–1^, respectively. The nutrient solution was pumped from the storage tank below the plants with a flow rate of 4 L h^–1^ and using a drip irrigation system with one emitter per plant. The nutrient solution was circulated continuously and up to 85 L was supplemented weekly with freshly prepared nutrient solution. 

The salt treatments were initiated 34 days after transplanting by addition of 1168.75 mg·L^–1^ NaCl (i.e., 20 mM NaCl) and 2337.50 mg·L^–1^ (i.e., 40 mM NaCl) to the nutrient solution. The electrical conductivity and pH of the treated nutrient solutions for 20 mM NaCl were 4.7 dS·m^–1^ and 5.8 dS·m^–1^, respectively, and for 40 mM NaCl, 7.3 dS·m^−1^ and 5.9 dS·m^–1^, respectively. The salinity stress was maintained for 104 days—from day 34 to day 138 after transplanting (26 June to 11 October 2018). The electrical conductivity and pH of the nutrient solution were measured at each stage of preparation of the fresh solution using a hand-held electrical conductivity and pH measuring device (CyberScan PCD650; Eutech Instruments, Thermo Fisher Scientific, Eutech Instruments Pte Ltd, Keppel Logistic Building, Singapore)). The pH of each nutrient solution was adjusted from 5.8 to 6.1 by adding 74% H_2_SO_4_. The mean air temperature and relative humidity were recorded inside the greenhouse every hour using a USB data logger (DL-120TH; Voltcraft, Hirschau, Germany). The mean daytime temperature was between 16 °C and 27 °C and never dropped below 13 °C during the night. The relative humidity was between 60% and 98%, and the daily maximum photosynthetically active radiation inside the greenhouse directly from the sunlight ranged from 800 µmol·m^–2^·s^–1^ to 1480 µmol·m^–2^·s^–1^, as measured using a spectrometer (LI 850; LI-COR Biosciences, Homburg, Germany).

### 4.2. Leaf and Fruit Sampling

The fruit harvest began 63 days after the transplantation. Fully developed yellow-green fruit samples were harvested every 8 to 10 days; there were a total of eight pickings, where the marketable and nonmarketable fruit samples were weighed. The nonmarketable fruit samples were mainly damaged by blossom end rot and were also measured and expressed as the proportion of the nonmarketable yield (%). At the end of the experiment, the biomass of the upper part of the plants was also determined. On each sampling date (27 July 2018, 16 August 2018, and 9 September 2018) two leaves and two fruit from each plant were randomly selected for analysis (i.e., proline, photosynthetic pigments, minerals, Cl– analysis), including the chemical analysis (i.e., total sugars, organic acids, ascorbic acid, total phenolics), for each condition of salinity and graft combination, and from each of three replications.

The yield data and the physiological and biochemical parameters were obtained by measurements and analysis of leaf and fruit samples taken from nine plants for each salinity treatment and grafting combination (i.e., ungrafted/grafted plants), as there were three plants per replicate and three replicates of the salinity/grafting combination (experimental unit, n = 3).

The analysis of proline and Cl^–^ in the leaves, and sugars, organic acids, and phenolics in the fruit were performed on fresh samples, while the photosynthetic pigments and minerals were measured in dry material. 

For analysis of the photosynthetic pigments, the bell pepper plant leaves were placed in a plastic bag and transported to the laboratory in a portable refrigerator. They were then frozen in liquid nitrogen, lyophilized, ground to a fine powder, and stored in brown, moisture-proof plastic containers at −80 °C until analysis. For the dry weight, 5 g frozen sample was freeze-dried for 22 h in a lyophilizer (Gamma 2–20; Christ, Osterode am Harz, Germany), and the water content (%) was calculated from the difference between the mass before and after lyophilization. 

### 4.3. Proline Determination

At each sampling date, fully expanded healthy leaves from the pepper plants were sampled (usually the third, fourth, and fifth leaves from the shoot tip). The proline content was determined as described by Bates et al. [63], with slight modifications. Fresh pepper leaf tissue (1 g) was ground in a mortar after the addition of a small amount of quartz sand and 10 mL 3% *(w/v)* aqueous sulfosalicylic acid. The homogenate was then centrifuged at 11,000× g for 12 min at 4 °C, and 1 mL of the supernatant was mixed with glacial acetic acid and ninhydrin reagent (1 mL each). The tubes with the reaction mixture were kept in boiling water for 1 h, and the reaction was stopped by putting the tubes in ice for 10 min. The absorbance was immediately measured at 546 nm in a spectrophotometer (U*v*/*v*IS Lambada Bio20; Perkin Elmer, Fullerton. Canada). The proline concentrations were calculated from the standard curve plotted against known concentrations of L-proline (Fluka, BioChemica, Switzerland), and are expressed as mg·g^–1^ fresh weight (FW).

### 4.4. Photosynthetic Measurements

Measurements of the net photosynthesis (Pn), stomatal conductance (g*_s_*), and transpiration (*E*) in the pepper plants were carried out from 9:00 to 12:00 using a portable measuring system (LI-6400; Li-Cor Bioscience, Lincoln, NE, USA) equipped with a fluorometer. Young, fully expanded leaves were trapped in the chamber and the measurements were made under the external CO2 concentration of 400 µmol·mol^–1^ and saturation light intensity, with a photosynthetic photon flux density of 1000 µmol·m^–2^·s^–1^. The temperature and air humidity were set to the mean daily values for the particular measuring day/period. The midday leaf water potential (Ψ) was measured (Scholander pressure chamber 3005-1223, Soil Moisture Equipment Corp., Goleta, CA, USA) on the same leaf after the leaf gas exchange measurement, (from 12:00 to 13:00), to determine the water status of the plant. After the physiological measurements were complete, the relative water content was measured for the leaves using the standard method [64]. 

### 4.5. Analysis of Photosynthetic Pigments

Extraction of the photosynthetic pigments from 150 mg dry powdered leaf material was performed with 3 mL 100% ice-cold acetone under dimmed light, as described by Sircelj et al. (2007) [65]. The samples were homogenized (Ultra Turrax T-25; IKA, Labotechnik, Stauden, Germany) for 30 s, filtered through a polyamide filter (Minisart SRP 15, polytetrafluoroethylene; Sartorius Stedim Bitech, Germany), and transferred to brown vials. An HPLC system with a photodiode array detector (Thermo Finnigan, San Jose, CA, USA) adjusted to 440 nm and equipped with a Spherisorb column (S5 ODS-2; 250 × 4.6 mm; Alltech Associates Inc., Deerfield, IL, USA) was used for the analysis of the compounds. The flow rate was maintained at 1 mL·min–1, with a running time of 32 min. Mobile phase A was HPLC grade acetonitrile, bidistilled water, and methanol (100/10/5; *v*/*v*/*v*), and mobile phase B was acetone with ethyl acetate (2/1; *v*/*v*). The solvent gradient was as follows (%B): 0→18 min, 10%→75%; 18→25 min, 75%→70%; 25→30 min, 70%→100%; 30→32 min, 100%→10%. 

Identification of the photosynthetic pigments was performed with the corresponding external standards based on their retention times and the photodiode array spectra. All of the measurements were performed in duplicate. Quantification of the compounds identified was performed on the basis of the peak areas and is expressed in µg·g^–1^ dry weight (DW) of the samples.

### 4.6. Analysis of Chloride Content

The chloride content was determined using the Mohr titration method [66]. For the extraction, 10 g fresh leaves were homogenized for 2 min in 10 mL double-distilled water (Ultra Turrax T-25; IKA, Labotechnik, Stauden, Germany). The homogenate was centrifuged at 8000 rpm for 10 min. Ten milliliters supernatant was then mixed with 2 mL 5% K_2_CrO_4_ and titrated with 0.1 M AgNO_3_. The equilibration point was reached when the color turned carmine red. Here, 0.01 M AgNO_3_ was used for titration of the samples from the control plants. The chloride content was calculated using the Mohr equation (Equation (1); [66]), and the data are expressed in mg·g^–1^ DW.
(1)$$\%\mathrm{Chloride}=\frac{\mathrm{mL}\;AgNO3}{g\;\mathrm{sample}}\times \frac{\mathrm{mol}\;AgNO3}{L}\frac{35.5\;g\;\mathrm{Cl}}{\mathrm{mol}\;\mathrm{NaCl}}\times \frac{1L}{1000\mathrm{mL}}\times 100\times \;\mathrm{dilution}\;\mathrm{factor}$$

### 4.7. Analysis of Mineral Content

A microwave digestion system (Ethos UP) with a −15 SK rotor was used for sample digestion. The digestion was performed in 100-mL polytetrafluoroethylene containers. All of the instruments were soaked overnight with 10% nitric acid and then rinsed with double-distilled water before use. The lyophilized plant samples (0.20 g) were weighed into the polytetrafluoroethylene containers. Two milliliters of 30% aqueous (*v*/*v*) hydrogen peroxide (Suprapur, Merck) and 8.0 mL 65% aqueous (*v*/*v*) nitric acid (Suprapur, Merck) were added to the containers. At least one reagent blank solution was prepared in the same way for each digestion kit. The samples were digested at the full power of 1800 W, as a ramp from 0→15 min to 180 °C, then15→25 min at 180 °C, followed by 15 min cooling. The digested solutions were diluted to 50 mL with double-deionized water (resistivity, 18.2 MΩ; Millipore). The contents of the elements were determined by mass spectrometry (ICP-MS 7900; Agilent, Tokyo, Japan) with an octopole reaction system. Sample introduction consisted of a quartz double-pass spray chamber and nebulizer (Agilent, Santa Clara California, United States) connected to the peristaltic pump of the spectrometer (Tygon tubes). Nickel samplers and skimmer cones were used. The operating conditions for the mass spectrometry were: RF power, 1550 W; plasma gas, 15 L·min^–1^; auxiliary gas, 0.9 L·min^–1^; carrier gas, 1.0 L·min^–1^. Helium (flow rate, 5.0 mL·min^–1^) was used as the reaction gas. The monitored isotopes were 23Na, 24Mg, 27Al, 39K, 43Ca, 51V, and 52Cr. The internal standards used were 45Sc, 115In, and 159Tb. The quantitative determination of the elements in the samples was performed using calibration curves obtained from diluted multi-element standard samples (TraceCERT, Sigma-Aldrich). Certified reference material (NIST SRM 1573a Tomato Leaves; Gaithersburg, MD, USA) was used to determine the accuracy of the results. All of the data are given as mg·g^–1^ DW.

### 4.8. Analysis of Sugars and Organic Acids

The pericarp tissue of the pepper fruit was analyzed for contents of glucose, fructose, sucrose, malic acid, and citric acid. For the extraction of the individual sugars and organic acids, 3 g FW of each sample was homogenized in 15 mL double-distilled water (Ultra Turrax T-25; IKA, Labotechnik, Stauden, Germany). The samples were left at room temperature for 30 min, with frequent stirring. The homogenate was then centrifuged at 10,000× *g* rpm for 5 min at 4 °C. The supernatants were filtered through cellulose filters (Chromafil A 20/25; Macherey-Nagel, Dueren, Germany), and transferred into vials, with 20 µL used for the analyses. The analyses for sucrose, glucose, fructose, malic acid, and citric acid were performed by HPLC (Finnigan Surveyor HPLC system; Thermo Scientific, San Jose, CA, USA). The chromatographic conditions were as described by Mikulic-Petkovsek et al. [67]. The carbohydrates and organic acids were calculated using an appropriate external standard. The concentrations are expressed as g·kg^–1^ FW.

### 4.9. Analysis of Ascorbic Acid

For ascorbic acid, the pericarp tissue of pepper fruit was chopped into small pieces with a ceramic knife, and 2.5 g crushed pericarp tissue was immediately mixed with 5 mL 2% metaphosphoric acid and ground thoroughly in a ceramic mortar, as reported by Mikulic-Petkovsek et al. [68]. The samples were left on a shaker for 30 min and then centrifuged at 9000× *g* rpm for 7 min at 4 °C (5801R; Eppendorf, Hamburg, Germany). The supernatants were filtered through cellulose filters (Chromafil A-20/25; Macherey-Nagel, Dueren, Hamburg, Germany), transferred to vials, and analyzed by HPLC (Finnigan spectra system; Thermo Scientific, Waltham, MA, USA), as previously reported [69]. The ascorbic acid concentrations were determined using the calibration curves established using an appropriate external standard, and the data are expressed as mg·(100 g)^–1^ FW.

### 4.10. Analysis and HPLC-MS^n^ Identification of Phenolic Compounds

Extraction of the pepper fruit pericarp was performed as described above and by Mikulic-Petkovsek et al. (2013) [68], with slight modifications. Fresh tissue samples were cut into small pieces and ground in a mortar cooled with liquid nitrogen. Then 5 g was extracted with 8 mL methanol with 3% (*w*/*v*) formic acid in an ultrasonic ice bath for 1 h. After extraction, the samples were centrifuged at 9000× *g* rpm for 7 min and then filtered through 0.20-µm polyamide filters (Chromafil AO-20/25; Macherey-Nagel, Dueren, Hamburg, Germany) and transferred to vials before injection into the HPLC system.

The individual phenolics were analyzed on the HPLC system (Accela; Thermo Scientific, San Jose, CA, USA) with a diode detector, at 280 nm (flavones), 310 nm (hydroxycinnamic acid derivatives), and 350 nm (flavonols), under the conditions described by Wang et al. (2002) [70]. All of the phenolic compounds were identified by mass spectrometry (LCQ Deca XP MAX; Thermo Scientific) with electrospray ionization and operated in negative ion mode, as described by Mikulic-Petkovsek et al. [69].

The concentrations of phenolic compounds were calculated from peak areas of the sample and the corresponding standards and were expressed as mg·(100 g)^−1^ FW.

### 4.11. Statistical Analysis

The effects of three factors were examined here—salt stress (salinity; S), at two NaCl concentrations (20 mM and 40 mM) and 0 mM NaCl control; grafting (G) at two levels (ungrafted, grafted); and date of sampling/measurements (D) (i.e., duration of treatment). For outcome variables of photosynthetic parameters, Cl^−^ leaf content, and fruit biochemical profile, the date had three durations (July, August, September), for proline and photosynthetic pigments, two duration (July, September), and for mineral leaf content and yield, one duration (September). 

Three tanks were used for each salt stress level, and for each tank, there were three ungrafted and three grafted plants. Considering that three plants for the same treatment were grown in the same tank (experimental unit), the measurements/analyses on these three plants were calculated as the means for the statistical analysis. Altogether there were nine values (three salt treatment × three replication) per group of plants (ungrafted/grafted) for each sampling/measurement date.

The main objective was to compare the ungrafted and grafted plants for each salt stress and date combination. As the measurements were carried out over time on the same plants within the tank, repeated measurement ANOVA was suitable for the statistical analysis. For each outcome variable, a linear mixed model was analyzed, with comparisons between “ungrafted” and “grafted” for each salt stress and date combination. Multiple comparison analysis at a significance level of *p* ≤ 0.05 was performed to determine the significances of the differences between the means, as required. 

The data were analyzed with the “*ggplot2*,” “nlme,” and “multcomp” packages in the R software, version 3.2.6 [71].

## 5. Conclusions

The salt-tolerant rootstock “Rocal F1” used in this study had no effect on scion physiology under salt stress. Similar water potential decreases and stomatal limitations of photosynthesis in grafted and ungrafted plants indicate that grafting did not mitigate the disturbances in water balance under increased salt exposure. In addition, the rootstock did not affect the uptake and distribution of Na^+^ and Cl^−^. Consequently, growth and yield reductions were not dependent on grafting.

Changes in fruit biochemical profile at higher salt stress (40 mM NaCl) indicated rootstock mediation through changes in functional compounds, but only in the early treatments in July, as evidenced by increased flavonoid content in fruit of grafted pepper plants (versus ungrafted). As the duration of salt stress increased, the ascorbic acid content increased, but without any influence of the rootstock. 

We suggest that the lack of alleviating effect of the grafting rootstock might result from the growth conditions. The interactions of Na^+^ and Cl^−^ with the mineral nutrients are of particular importance. We suggest further experiments with different supplies of essential minerals (reduced amounts of potassium, calcium, etc.) and different levels of salinity, which would provide better insight into the potential and limits of salt tolerance of rootstocks. 

## Figures and Tables

**Figure 1 plants-10-00314-f001:**
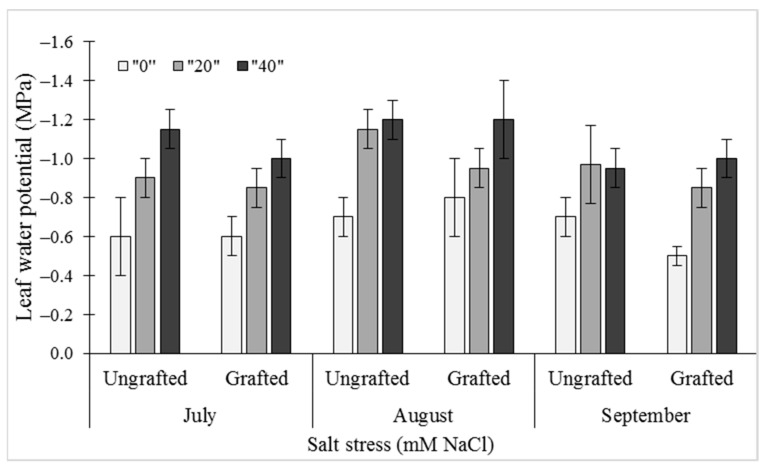
Effects of salt stress (0 mM, 20 mM, 40 mM NaCl) on midday leaf water potential (ψ) in leaves of ungrafted UG) and grafted (G) bell pepper plants, as measured in July, August, and September. Data are means ± standard deviation of three replicate samples (n = 3).

**Figure 2 plants-10-00314-f002:**
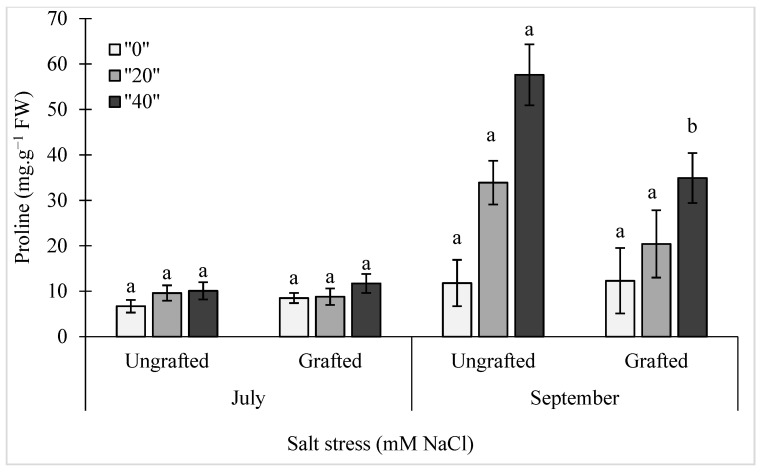
Effects of salt stress (0 mM, 20 mM, 40 mM NaCl) on proline content in the leaves of ungrafted and grafted bell pepper plants, as measured in July and September. Data are means ± standard deviation of three replicate samples (n = 3). Data with different lower case letters are significantly different (Duncan test).

**Table 1 plants-10-00314-t001:** Effects of salt stress on marketable yield, blossom end rot infection, and plant biomass for ungrafted and grafted pepper plants, and their salinity and grafting interactions.

Salinity (mM NaCl)	Marketable Yield (kg Plant^–1^)		Blossom End Rot Infection (%)		Plant Biomass (kg)	
	Ungrafted	Grafted	Ungrafted	Grafted	Ungrafted	Grafted
0	1.34 ± 0.2	1.12 ± 0.1	25.5 ± 5 ***	42.2 ± 5	0.970 ± 0.20	0.78 ± 0.06
20	0.96 ± 0.3	0.84 ± 0.1	65.9 ± 11	65.6 ± 11	0.777 ± 0.03	0.63 ± 0.02
40	0.64 ± 0.3	0.51 ± 0.1	72.5 ± 8 ***	52.0 ± 8	0.577 ± 0.02	0.44 ± 0.07
**Significance**						
Salinity (S)	*		*		*	
Grafting (G)	ns		ns		ns	
S × G	ns		**		ns	

Data are means ±standard error (n = 3). Symbols indicate statistically significant differences between ungrafted and grafted plants. *, *p* < 0.05; **, *p* < 0.01; ***, *p* < 0.001; ns, not significant (ANOVA).

**Table 2 plants-10-00314-t002:** Effects of salt stress on leaf photosynthetic traits in July, August, and September for ungrafted and grafted pepper plants, and their salinity, grafting, and date interactions.

Salinity	Grafting	Net CO_2_ Assimilation Rate (P_n_)			Stomatal Conductance (g_s_)			Transpiration Rate (E)		
(mM NaCl)		July	August	September	July	August	September	July	August	September
0	Ungrafted	20.3 ± 1.1	14.4 ± 1.9	17.7 ± 0.6	0.69 ± 0.12	0.25 ± 0.04	0.50 ± 0.02	5.83 ± 0.55	3.91 ± 0.74	3.28 ± 0.43
	Grafted	17.4 ± 2.2	12.0 ± 0.6	18.0 ± 0.6	0.53 ± 0.2	0.22 ± 0.04	0.61 ± 0.03	5.05 ± 1.42	3.80 ± 0.57	3.50 ± 0.40
20	Ungrafted	18.0 ± 0.8	11.3 ± 0.8	15.2 ± 2.3	0.51 ± 0.12	0.15 ± 0.02	0.42 ± 0.09	5.45 ± 0.70	2.82 ± 0.33	2.72 ± 0.45
	Grafted	20.5 ± 1.6	15.6 ± 0.6	18.1 ± 0.9	0.76 ± 0.09	0.27 ± 0.04	0.66 ± 0.08	6.92 ± 0.50	4.70 ± 0.64	3.87 ± 0.41
40	Ungrafted	18.2 ± 1.7	8.6 ± 0.6	9.0 ± 2.4	0.54 ± 0.14	0.12 ± 0.01	0.21 ± 0.05	5.19 ± 0.97	2.10 ± 0.31	1.76 ± 0.14
	Grafted	17.3 ± 3.5	12.1 ± 2.6	12.2 ± 2.3	0.45 ± 0.13	0.17 ± 0.05	0.27 ± 0.05	4.87 ± 1.12	3.36 ± 0.97	2.10 ± 0.47
**Significance**										
Salinity (S)		*			*			ns		
Grafting (G)		ns			ns			ns		
Date (D)		***			***			***		
S × G		*			ns			ns		
S × D		ns			ns			ns		
G × D		ns			ns			ns		
S × G × D		ns			ns			ns		

Data are means ± standard errors (n = 3). Symbols indicate significant differences between treatments; *, *p <* 0.05; ***, *p* < 0.001; ns, not significant (ANOVA).

**Table 3 plants-10-00314-t003:** Effects of salt stress on further leaf photosynthetic traits measured in July, August and September for ungrafted and grafted bell pepper plants, and their salinity, grafting, and date interactions.

Salinity (mM NaCl)	Grafting	Maximum Quantum Use Efficiency of PSII in Dark-Adapted State (Fv/Fm),			Effective Quantum Use Efficiency of PSII in Light-Adapted State (Fv′/Fm′)			Photochemical Quenching (q_P_)		
		July	August	September	July	August	September	July	August	September
0	Grafted	0.76 ± 0.00	0.76 ± 0.00	0.70 ± 0.06	0.53 ± 0.02	0.51 ± 0.02	0.57 ± 0.02	0.45 ± 0.02	0.38 ± 0.02	0.41 ± 0.01
	Ungrafted	0.76 ± 0.00	0.76 ± 0.00	0.76 ± 0.00	0.56 ± 0.02	0.52 ± 0.03	0.55 ± 0.02	0.48 ± 0.02	0.43 ± 0.04	0.41 ± 0.01
20	Grafted	0.76 ± 0.00	0.76 ± 0.00	0.76 ± 0.00	0.56 ± 0.01	0.53 ± 0.01	0.57 ± 0.00	0.49 ± 0.02	0.46 ± 0.01	0.37 ± 0.00
	Ungrafted	0.76 ± 0.00	0.76 ± 0.00	0.72 ± 0.04	0.54 ± 0.02	0.48 ± 0.01	0.54 ± 0.04	0.45 ± 00.02	0.43 ± 0.03	0.39 ± 0.01
40	Grafted	0.76 ± 0.00	0.76 ± 0.00	0.71 ± 0.06	0.52 ± 0.04	0.49 ± 0.02	0.49 ± 0.03	0.46 ± 0.02	0.42 ± 0.00	0.40 ± 0.02
	Ungrafted	0.76 ± 0.00	0.76 ± 0.00	0.71 ± 0.05	0.53 ± 0.02	0.45 ± 0.02	0.46 ± 0.05	0.47 ± 0.01	0.43 ± 0.00	0.41 ± 0.00
**Significance**										
Salinity (S)		ns			*			ns		
Grafting (G)		ns			ns			ns		
Date (D)		*			***			***		
S × G		ns			ns			ns		
S × D		ns			ns			ns		
G × D		ns			ns			ns		
S × G × D		ns			ns			ns		

Data are means ± standard errors (n = 3). Symbols indicate significant differences between treatments; *, *p <* 0.05; ***, *p* < 0.001; ns, not significant (ANOVA).

**Table 4 plants-10-00314-t004:** Effects of salt stress on photosynthetic pigment characteristics and xanthophyll cycle components measured in July and September for ungrafted and grafted bell pepper plants, and their salinity, grafting, and date interactions.

Salinity (mM NaCl)	Grafting	Total Chlorophyll Content (Chl_a+b_)		Total Violaxanthin, Antheraxanthin, and Zeaxanthin		De-Epoxidation State (EOS)	
		July	September	July	September	July	September
0	Ungrafted	10.98 ± 0.22	10.88 ± 0.10	0.52 ± 0.01	0.57 ± 0.01	0.57 ± 0.01	0.18 ± 0.02
	Grafted	11.02 ± 0.21	10.77 ± 0.11	0.55 ± 0.03	0.54 ± 0.03	0.54 ± 0.03	0.13 ± 0.03
20	Ungrafted	10.62 ± 0.42	10.42 ± 0.13	0.50 ± 0.03	0.51 ± 0.03	0.51 ± 0.03	0.14 ± 0.03
	Grafted	10.78 ± 0.02	10.74 ± 0.12	0.50 ± 0.01	0.53 ± 0.02	0.53 ± 0.02	0.11 ± 0.01
40	Ungrafted	10.06 ± 0.22	9.59 ± 0.39	0.42 ± 0.01	0.43 ± 0.03	0.43 ± 0.03	0.11 ± 0.01
	Grafted	10.79 ± 0.51	10.24 ± 0.13	0.46 ± 0.03	0.43 ± 0.02	0.43 ± 0.02	0.13 ± 0.01
**Significance**							
Salinity (S)		ns		**		ns	
Grafting (G)		*		ns		ns	
Date (D)		**		ns		***	
S × G		ns		ns		ns	
S × D		ns		ns		ns	
G × D		ns		ns		ns	
S×G×D		ns		ns		ns	

Data are means ± standard errors (n = 3). Symbols indicate significant differences between treatments; *, *p <* 0.05; **, *p* < 0.01; ***, *p* < 0.001; ns, not significant (ANOVA).

**Table 5 plants-10-00314-t005:** Effects of salt stress on leaf mineral concentrations measured in September for ungrafted and grafted bell pepper plants, and their salinity and grafting interactions.

Salinity	Grafting	Leaf Mineral Concentration (mg·g^–1^ Fresh Weight)				
(mM NaCl)		Na^+^	Cl^–^	K^+^	Ca^2+^	Mg^2+^
	Ungrafted	0.014 ± 0.01	0.17 ± 0.052	61.5 ± 7.38	20.2 ± 4.90	10.2 ± 1.11
0	Grafted	0.016 ± 0.01	0.15 ± 0.042	51.5 ± 6.01	20.6 ± 5.17	10.0 ± 1.48
	Ungrafted	0.557 ± 0.35	2.84 ± 0.563	54.4 ± 5.40	21.1 ± 4.58	10.4 ± 2.00
20	Grafted	0.291 ± 0.09	2.98 ± 0.558	54.8 ± 7.96	23.3 ± 8.35	11.5 ± 2.87
	Ungrafted	1.190 ± 0.61	4.73 ± 0.414	56.4 ± 8.26	28.9 ± 5.98	11.8 ± 1.01
40	Grafted	0.955 ± 0.35	5.61 ± 0.622 ***	52.8 ± 4.53	27.6 ± 6.98	12.0 ± 2.20
**Significance**						
Salinity (S)		***	***	ns	*	*
Grafting (G)		ns	ns	ns	ns	ns
S × G		ns	*	ns	ns	ns

Data are means ±standard errors (n = 3). Symbols indicate significant differences between treatments; *, *p <* 0.05; ***, *p* < 0.001; ns, not significant (ANOVA).

**Table 6 plants-10-00314-t006:** Effects of salt stress on the biochemical parameters of the pepper fruit harvested in July and September for ungrafted and grafted bell pepper plants, and their salinity, grafting, and date interactions.

Salinity	Grafting	Total Sugars		Total Organic Acids		Ascorbic Acid		Total Phenolics Analysed	
(mM NaCl)		(g·kg^–1^ Fresh Weight)		(g·kg^–1^ Fresh Weight)		(mg·[100 g]^–1^ Fresh Weight)		(mg·[100 g]^–1^ Fresh Weight)	
		July	September	July	September	July	September	July	September
0	Grafted	2790 ± 438	2460 ± 447	226 ± 27	192 ± 23	50.5 ± 28	57.4 ± 27	5.96 ± 2.6	4.47 ± 1.4
	Ungrafted	2710 ± 638	2630 ± 370	230 ± 41	203 ± 42	34.8 ± 15	57.8 ± 25	6.43 ± 2.1	3.34 ± 1.7
20	Grafted	2680 ± 324	3530 ± NA	220 ± 35	245 ± NA	44.9 ± 19	82.9 ± NA	5.24 ± 0.8	1.8 ± NA
	Ungrafted	2990 ± 280	2940 ± 197	252 ± 34	317 ± 73	51.2 ± 24	82.9 ± 14	5.84 ± 1.3	2.87 ± 0.9
40	Grafted	2760 ± 359	2690 ± 1090	249 ± 24	371 ± 50	57.4 ± 19	92.6 ± 9	6.77 ± 1.2 *	2.99 ± 0.5
	Ungrafted	2970 ± 559	3480 ± 306	235 ± 18	415 ± 47	49.9 ± 16	102 ± 13	4.85 ± 1.1	3.15 ± 0.4
**Significance**									
Salinity (S)		Ns		***		**		ns	
Grafting (G)		Ns		*		ns		ns	
Date (D)		Ns		***		***		***	
S × G		Ns		*		ns		ns	
S × D		Ns		***		**		ns	
G × D		Ns		ns		ns		ns	
S × G × D		Ns		ns		ns		*	

Data are means ±standard error (n = 3); NA, not available (n = 1). Symbols indicate significant differences between treatments; *, *p <* 0.05; **, *p* < 0.01; ***, *p* < 0.001; ns, not significant (ANOVA).

## Data Availability

All data pertaining to this study is being held in computers owned by University of Ljubljana, Ljubljana, Slovenia, under control of the PI team.

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
