# Peer review of "Physiological and Biochemical Responses of Ungrafted and Grafted Bell Pepper Plants (Capsicum annuum L. var. grossum (L.) Sendtn.) Grown under Moderate Salt Stress"

_plants, 2021, doi:10.3390/plants10020314_

Round 1

Reviewer 1 Report

Dear Authors

Present manuscript entitled "Physiological and Biochemical Parameters of Ungrafted and Grafted Bell Pepper Plants (Capsicum Annuum L. var. Grossum (L.) Sendtn.) Grown Under Moderate Salt Stress" demonstrated the bell pepper variety ’Vedrana’ grafted onto the salt-tolerant root stock ‘Rocal F1’ and grown at three salinities. Please find here some of queries-

1- Please explain the significance of moderate salt stress and selection of three treatments (0, 20, 40 mM NaCl). Actually there are two salinity concentrations and 0 represent the control. 

2- Introduction is very descriptive and in details of general information, it may be reframed to more compact and small. 

3- Line 43-46. Sweet pepper (Capsicum annuum L.) is a widespread and popular crop. When grown in arid and semi-arid regions it is often associated with salt problems, and the plants are considered to be sensitive to moderate
 soil salinity. What does it mean by salt problems? Please reframe the sentence.

4- Results are very well presented and discussed in details while methods are described nicely in details. However conclusion is again in much detail in my opinion. It is better to draw a short conclusion with clear message of the study, which may facilitate the readers to understand in better way.

Thank you

Regards

Author Response

Dear Reviewer #1,

Reviewer 2 Report

The author investigated the physiological and biochemical responses of grafted and un-grafted bell pepper under two doses of salinity.

The manuscript in the current forms has a lot of shortcomings and according to my opinion they have methodological problems which turned to many non-significant results. Due to the following major reasons, the authors need to precisely design an experiment and consider the following points to avoid confusion for the readers and provide concrete results and discussion.

Major points:

  1. Plants have grown in hydroponic condition. The nutritional solution has been considered wrongly. When we deal with salt stress, we must not provide huge amount of K, Mg and Ca for plants otherwise we cannot induce salt stress. He the authors used 6 mM K and 4 mM Ca which is super luxury for plants. Author did not consider that cations will compete hardly together. That is why the did not see significant differences withing most of the provided results.
  2. The authors have chosen two doses of salinity for this study. 20 and 40 Mm. There is not data to support that salt stress has been induced. Also, it is not clear which phase of salinity has been considered, Ionic or toxic? First, you need to show and test if 20 or 40 mM NaCl is enough for your study and then apply for the main experiment. Why not 50 or 60 mM?
  3. When we deal with salt stress, potassium plays very important roles. In line 184 the authors mentioned that ‘’Although the K+ levels in the leaves of these un-grafted and grafted plants indicated similar salt stress responses’’… without considering the fact that how plants can response to Na stress with huge amount of provided K? Here is the point that author chose wrongly the national solution.

  1. In the same lines (184-197) authors avoid explaining why K is not changing and instead they highlighted that Cl plays role and that is why they has yield reduction. The question is if the scope of the study is Cl toxicity? How do you sure that you are in the toxicity level of Cl? Which treatments did you includ? Which parameters you provided?

  1. Also, another crucial point is that authors reported too many non-significant results. Hence, tthe question arise why authors reported these results? I could not find the effect of grafting on fruit yield, water relation, photosynthesis, minerals, biochemical analysis (except total phenolic compound).  

  1. Author also explained their results only based on ANOVA. ANOVA will be very good idea if we would like to consider the interaction between different factors and to find which one influencing more. But this must apply after a valid statistical analysis. Most of the provided data did not show statistically significant and authors frequently highlighted this fact that the analysis are not significant. Only Blossom end rot infection, proline, Cl and total phenolic compound are significant which is not sufficient for a paper.

Minor points:

The English is poor and must be extensively reviewed.

Better not using parameters in the tile. Parameters is for results section. Here the term responses need to be replaces.

Introduction need be carefully and precisely re-written. Very divers and far to understand. A lot of mistake. What does mean fight (line 63)?? I suggest considering better terms like cope, tolerate.

The results section needs to be attentively explained. Very difficult to understand. There are a lot of English and grammar problem. Example line 173 ‘’the grafting did not reach’’. We can use ‘’did not show any statistical differences’’ instead of reach. This problem exists in whole manuscript.

  • Where is salt stress marker to show that you induced salt stress in plants?
  • How do you sure 40 Mm Na was enough to induce salt stress in your condition?
  • There is no root data so many of the conclusions can not be considered. It must be provided.
  • How you explain the accumulation of Cl in the tissue? In line 188-189 you mentioned ‘’the inability of plant to limit the transport of Cl’’. Did you measure the elements in roots? Or any genes involved in Cl transporters to support it?
  • Regarding the statement of K on line 184, you need data to show that K was not too high. And to clearly show this is not the reason for low response of plants to salt stress.

Discussion is very poor and need to be re-written. Author need to explain the results in a good way and discuss the reasons carefully.  Here I can see only comparison with the other studies. When you compared with other papers better to use scientific terminology like in line with or in agreement with.

Well discussion regarding salt stress, proline, K, phenolic compounds need to be considered in Bell Pepper.

Author Response

Dear Reviewer #2,

Reviewer 3 Report

The manuscript ‘Physiological and Biochemical Parameters of Un-grafted and Grafted Bell Pepper Plants (Capsicum Annuum L. var. Grossum (L.) Sendtn.) Grown Under Moderate Salt Stress’ is scientifically sound and can be accepted for publication after minor modification in accordance to the given recommendations.

The manuscript shows very interesting data about grafting influence on bell pepper physiological and biochemical properties under long term salt stress conditions. Data about the influence of rootstock and salt conditions on fruit yield and quality are particularly interesting.

All comments are written at 'SPECIFIC COMMENTS’.

SPECIFIC COMMENTS

Introduction

Line 87 –  Please add another reference  …been assessed in a few studies [21, xy, xy…].

Materials and methods

Lines 306-309 – Is it means that you had ungrafted, self-grafted and grafted onto rootstock ‘Rocal’ ?

Line 315 -in 84-cell polystyrene trays … Please, add cell volume.

Lines 321 - 322 -   .. tubular silicone clip.  please add producer and clips hole diameter.

Lines 322 – Please,  describe the conditions during the formation of callus (T, RH, lighting).

Lines 322-323 – Therefore, the bell pepper seedlings consisted of ungrafted ‘Vedrana’ (control) and ‘Vedrana’ grafted onto the ‘Rocal’ rootstock.- This is said in sentences 306-309.

please check Optifort and Dynafort producer (De Ruiter - Bayer)

Line 428 - .. by Šircelj et al. (2007). – Please, add a reference to the literature and assign it a

Number

Line 515 - .. reported [6]… or 63? Please check reference.

Line 531- … Wang et al. (2002). – Please, add a reference to the literature and assign it a

Number

Line 560- Please change symbol ..<.. to ≤

Author Response

Dear Reviewer #3,

Round 2

Reviewer 2 Report

The comments has been transferred to the Editor